# How does subjective social status at school at the age of 15 affect the risk of depressive symptoms at the ages of 18, 21, and 28? A longitudinal study

**Marie Kjærgaard Lange**[1]*, **Vivi Just-Nørregaard**[1,2], **Trine Nøhr Winding**[1,2]

1 Department of Occupational Medicine–University Research Clinic, Danish Ramazzini Centre, Goedstrup Hospital, Herning, Denmark, 2 Department of Clinical Medicine, Faculty of Health, Aarhus University, Aarhus, Denmark

* marieklange@gmail.com

**Data Availability Statement:** Data from Statistics Denmark are used, and as these are subject to a license agreement, they are not publicly available. The data used during the current study are

## Abstract

### Background

Young people's mental health is declining. Depression is a public disease which is increasing internationally, and in Denmark an increase is seen especially among young people. Objective social status is known to be associated with mental health and depression, but little is known about the association between adolescent subjective social status at school and depressive symptoms during young adulthood. The aim was to investigate the association between 15-year-old's subjective social status at school and the development of depressive symptoms at age 18, 21 and 28.

### Methods

The study is a longitudinal study using questionnaire data from The West Jutland Cohort Study Denmark. The study population consisted of adolescents who at baseline, at age 15 (2004), had answered questions about their subjective social status in school using the MacArthur scale-youth version. Answers were categorised into low, medium, and high subjective social status. Outcome data about depressive symptoms was collected at age 18 (2007), age 21 (2010) and age 28 (2017) using the CES-DC and CES-D scales, dichotomised into few or many depressive symptoms. The associations between subjective social status at school at age 15 and depressive symptoms at ages 18, 21 and 28 were analysed using multiple logistic regression.

### Results

Statistically significant associations were found between low subjective social status at school at age 15 and the odds of many depressive symptoms at all three age points in young adulthood. When adjusting for co-variates the odds ratio for many depressive symptoms at age 18 was OR 3.34 [1.84;6.08], at age 21 OR 3.31 [1.75;6.26] and at age 28 OR 2.12 [1.13;3.97].

however available from the Statistic Denmark Institutional Data Access / Ethics Committee for researchers, who meet the criteria for access to confidential data upon request to the Department of Occupational Medicine, Goedstrup Hospital (contact via the Department of Occupational Medicine, email: arbejdsmedicin@goedstrup.rm.dk).

**Funding:** The author(s) received no specific funding for this work.

**Competing interests:** The authors have declared that no competing interests exist.

## Conclusions

The subjective social status of 15-year-olds is associated with depressive symptoms at ages 18, 21 and 28, respectively. It seems that subjective social status at age 15 is of greatest importance for the occurrence of depressive symptoms in the short run, and that the impact attenuates over time.

## Background

Good mental health is essential for maintaining overall good health as both physical and mental illness are affected by mental health [1]. Mental health can be assessed by the prevalence of mental disorders and their symptoms, such as anxiety and depression. WHO describes depression as a common emotional disorder which is increasing internationally [2]. The prevalence of Danes over age 16 who are depressed or are feeling depressed or unhappy have increased from 5.6% in 2010 to 8.3% in 2021 and the prevalence of mental health problems is also increasing among young people [3]. Similarly, a Danish report on mental health and illness in children and young adolescents shows that mental disorders such as anxiety and depression are especially increasing among 10 to 24-year-olds. Around 15% of Danish adolescents are diagnosed with a mental illness or other developmental disorder before the age of 18, especially children and adolescents with the lowest socioeconomic and social status are at risk. In early adolescence—the period after primary school—young people are particularly vulnerable to developing depression. This may be because, they now stand more on their own two feet and has to make decisions that affect their future life [4].

The transition from adolescence to adulthood is an important developmental life period in which many physical, psychological as well as social changes occur, and a large number of young people struggle with mental health problems in this particular life phase. It is well known that young people are particularly susceptible to these changes due to their not yet fully developed and still impressionable minds [5]. In addition, we know that mental problems in youth can have consequences for health later in life [6].

The causes of the negative tendency regarding mental health are complex and depend on biological, internal as well as external factors [3], like previous mental health problems [7, 8], bullying [9–11], poor family functioning during upbringing [12] and stress [13]. In addition, it is known that there are sex differences in the risk of developing depression [14, 15]. An association between high BMI or low social status, respectively, and mental health problems, including depressive symptoms is likewise well documented [16–18]

Social status is often assessed as objective socio-economic status (SES), which can be quantified through various methods [19]. In a Danish context, objective socio-economic status is often determined by factors such as income, educational level and/or labour market participation [12]. However, there is increased focus on the importance of subjective social status (SSS) as well.

SSS is an individual assessment of a person's own human, social and cultural capital and thereby specifically embraces the facets of social differences, privilege and marginalisation [20]. Several studies have found low SSS in adulthood to be linked to mental health conditions including depression and depressive symptoms [9, 12, 21–30]. For example, a meta-analysis by Quon and McGrath finds that SSS assessed by the MacArthur SSS Scale—Youth Version is strongly related to health outcomes related to psychological processes, including depression [31]. In young people SSS can be measured in two ways; either as SSS-society which is the self-

rated social status the young person assesses his/her family to have compared to other families in the society, or as SSS-school which is the young person's self-rated social status compared to the peers at school [26]. Glendinning et al thus found that the mutual influence of peers has a great influence on their self-perception and health behavior [32].

Social interactions are important for young people as they gauge their own status and value in comparison to their peers [33]. In Denmark, 98% of children attend either private or primary school, spending a substantial portion of their time with their schoolmates. Consequently, comprehending the attributes associated with youth social status requires an examination of the social relations at school [34].

It is likewise well-known that the social environment at school is associated with the mental health of young people, but it is mainly cross-sectional studies that describe the relationship [7, 9, 26, 27, 29, 30]. However, information about the impact of SSS-school on the development of depressive symptoms later in adolescence and in early adulthood and hence the consequence for mental health later in life is lacking. It is therefore relevant to investigate the association between low SSS-school and the development of depressive symptoms in Danish young adults.

The 3 ages outlined in the study represent important stages in the lives of Danish young people. At 18 years old, the culmination of puberty marks a detachment from the family, though a majority is still living at home. Around the age of 21, most have moved away from home. By the age of 28, most young adults have established their own residences with a steady job and perhaps a family [35].

SSS is reliable in predicting physical illness and mental health—including depression, [36, 37] and a study by Lemeshow et al. also shows that SSS among adolescents plays a causal role in shaping health and general wellbeing [37].

The aim of this this study is to investigate the association between SSS-school at age 15 and the risk of developing depressive symptoms at ages 18, 21 and 28.

## Methods

### Design and population

The study was a prospective cohort study that longitudinally aimed to investigate the association between 15-year-old's subjective social status at school and the development of depressive symptoms in adolescence and young adulthood.

The study was based on The West Jutland Cohort Study which contains questionnaire and register information on individuals born 1989 and living in the former Ringkøbing County, Denmark in 2004. The study was initiated in 2004 (age 15) and the participants have been followed with questionnaires continuously since. A more detailed and elaborate description of the cohort's methodology and recruitment is described elsewhere [38].

The source population for the study was 3681 individuals initially invited to participate. Of those a total of 3054 (83%) participated in the baseline questionnaire, from which the information on exposure, SSS-school, was obtained.

All participants from the source population were invited to participate at follow-up points in 2007, 2010 and 2017, from which information on the outcome, depressive symptoms, was obtained.

Participants were excluded from the analyses if they had not answered the question regarding subjective social status at school in 2004. Additionally, participants were excluded if they had not answered the questions regarding depressive symptoms at follow-up in either 2007, 2010 or 2017. Fig 1 illustrates the selection process of the participation in the study.

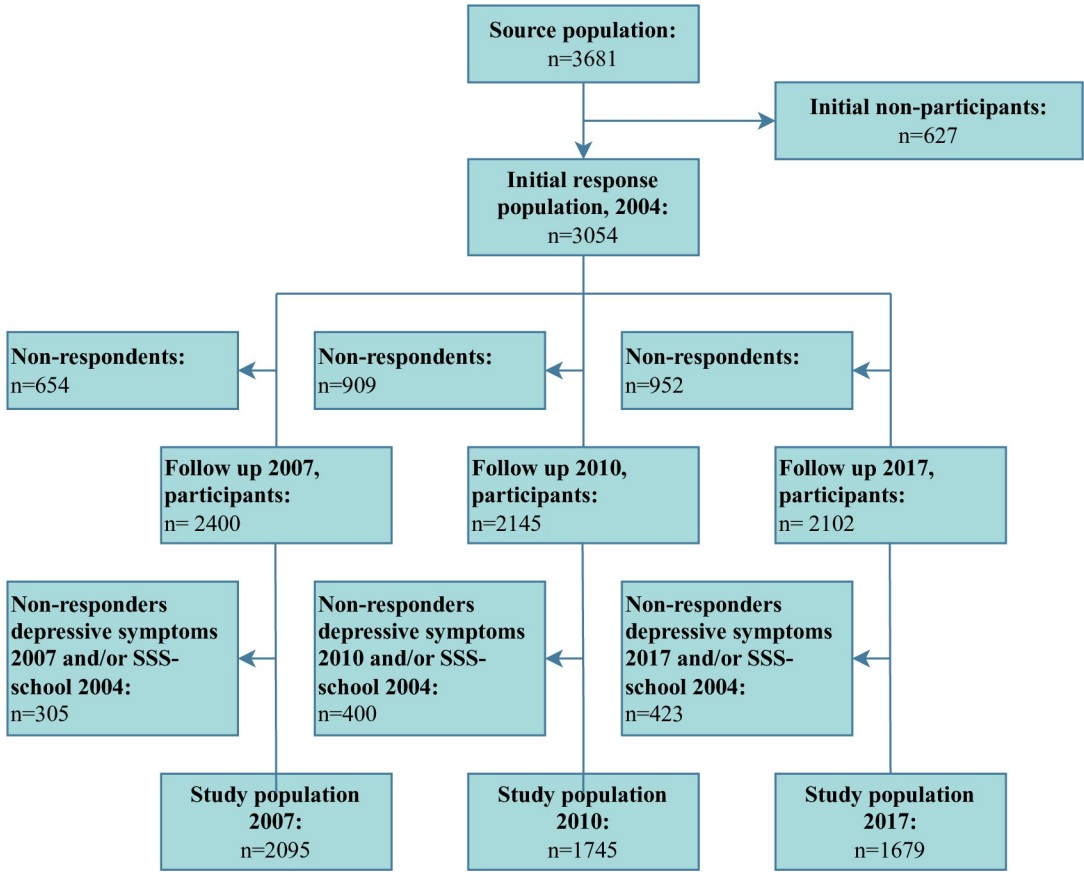

**Fig 1.**

## Data from registers

Register information about sex, household equivalised income and mothers educational level was extracted from the registers and linked to each study participant [39–41].

## Outcome

Depressive symptoms were measured by an overall question "During the past week, how much have you had the following feelings?" containing 4 items selected from the Center for Epidemiological Studies Depression Scale for Children (CES-DC) in 2007 and 2010, and the Center for Epidemiological Studies Depression Scale (CES-D) in 2017. The questions in the youth version were equivalent to the questions from the adult version but adapted to the context in which children and adolescents see themselves [42, 43]. Each item was a description of a feeling for which the respondent had to indicate the degree to which it applied to him/her, the four items were "I was happy", "I felt like kids I know were not friendly or that they didn't want to be with me", "I felt sad", and "It was hard to get started doing things". The degree of agreement could be indicated from "Not at all", "A little", "Sometimes" or "A lot" equivalent to a score between 0 and 3. The original scale, developed by Radloff et al. in 1977, contains 20 items with a range of 0–60 points. Scores of 15 and above are considered to show signs of depressive symptoms [44]. In this study, participants could score from 0–12 by answering the four questions. A cut point using the 75th percentile to differentiate between few and many depressive

symptoms was chosen as the population consist of predominantly healthy, young participants. The cut point used for the 2007 and 2010 study populations was thus ≥4, and ≥3 for 2017.

## Exposure

SSS-school is a measure that reflects the 15-year-old's subjective assessment of his/her own social status and position compared to peers in the class. SSS-school, in relation to peers, was measured using a Danish translated version of the MacArthur scale—youth version. The scale can contribute to the understanding of how adolescents subjectively assess their social status in social relations. By using the scale, you can provide knowledge and understanding of the impact of social status on health in both short and long term. The scale is a ten-step ladder on which the young person must place him/herself according to the following description [26]:

> *"Now assume that the ladder is a way of picturing your school. At the top of the ladder are the people in your school with the most respect, the highest grades, and the highest standing. At the bottom are the people whom no one respects, no one wants to hang around with, and who have the worst grades. Where would you place yourself on this ladder? Mark the rung that best represents where you would be on this ladder."*

In this study, SSS-school was categorised and divided into three categories consisting of the bottom three steps (low SSS-school), the middle four steps (medium SSS-school) and the top three steps (high SSS-school), allowing for the distinction between two groups of individuals who were distinctly less or more privileged when compared to a middle group. Additionally, this categorization aligns with other studies, making the comparison of results with existing research easier [12].

## Co-variates

Information about sex was obtained from the Danish Central Person Register (CPR) and consisted of the two categories men and women [45].

SSS-society was measured using the MacArthur scale—youth version, following the same categorisation as the SSS-school measure [26]. SSS-society measures the young person's subjective assessment of his/her own family's social status in society compared to other families in Denmark and should thus be seen as the family's status in society. Like SSS-school it was categorised into three: the bottom three steps (low SSS-society), the middle four steps (medium SSS-society) and the top three steps (high SSS-society). Information about SSS-society was collected when the participants were 15 years old.

Objective socioeconomic status was measured by the mean equivalised income and mothers' educational level. Equivalised household income is chosen as it is a weighted measure based on an OECD modified scale that takes into account household size and composition. Equivalised means that one adult in the household is counted in the calculation by a factor of 1.0, other adults and children over 14 by a factor of 0.5, and children (under 15) by a factor of 0.3. In addition, the annual household income was divided by the weighting factor each year [46].

In a Danish context, educational level and especially the mother's educational level has been found to have an impact on the future well-being of young people [12]. Finally, income was categorised at the 33.3 and 66.6 percentiles and hence categorised as lower, medium, or higher household income.

Information on the mother's highest completed education was assessed when the participants were 15 years old. Educational levels were categorised into four; < 10 years = primary/

elementary school, 10–13 years = secondary school, 13–15 years = bachelor's degrees, and > 15 years = master's and doctoral degrees.

Family functioning was presented as a score and hence the degree of how well the young person's family was functioning. This was measured subjectively at age 15 by 12 items from the McMaster Family Assessment Device (FAD) [47], by indicating the degree to which each statement was one's family with the response options '*Totally agree*', '*Mostly agree*', '*Mostly disagree*' or '*Strongly disagree*'. Each question allowed for 1 to 4 points, and an average score was calculated for each participant, with an average of 1 indicating the best possible family functioning and an average of 4 indicating the worst possible family functioning. The variable was dichotomized at the 75th percentile indicating either good family functioning (<2.08) or poor family functioning (≥2.08).

Bullying was assessed based on the individual's subjective experience at age 15, using the question "How much have you been bullied at school in the last six months?" with the possible answers: "Have not been bullied", "Once or twice", "Sometimes", "Once a week" or "Several times a week". The variable was categorized into three: None, mild ("Once or twice" and "Sometimes") and severe ("Once a week" and "Several times a week") bullying.

Depression [7, 8, 26], stress [8] and BMI [48] in adolescence were assessed as intermediating factors of the association between SSS-school and depressive symptoms later in life and therefore will not be adjusted for as confounders.

## Statistical methods

The distribution of characteristics of the participants was presented as number (n) and percentage (%) in relation to depressive symptoms at ages 18, 21 and 28.

Means were calculated for the level of depressive symptoms at ages 15, 18, 21 and 28. This was done for each of the three categories of SSS-school in order to generate trajectories of the development of depressive symptoms over time by grade of SSS-school.

Correlation analyses were performed between the exposure variable and co-variates using Spearman's rank correlation test.

The main analysis was performed by multiple logistic regression with calculation of Odds Ratio (OR) to estimate odds of many depressive symptoms at low, medium, and high SSS-school respectively. This was done for each of the three study populations 2007, 2010 and 2017. Crude as well as adjusted ORs were calculated with high SSS-school as reference group. Using multiple logistic regression, this study was able to examine the association between an exposure, SSS school, and a dichotomous outcome, depressive symptoms, as well as the influence of a number of co-variates on this association.

All odds ratios were presented with 95% confidence intervals (95% CIs) with a significance level of $p < .05$. The analysis software used was STATA statistical package 17.0 (StataCorp LLC, College Station, Texas, USA).

## Ethics

The study is approved by the Danish Data Protection Authority (Journal no.: 1-16-02-547-15), and as the cohort study uses data collected from questionnaire surveys and health science register data, the study is exempt from the requirement to notify the ethics committee according to the Committee Act § 14, paragraph 2 [49]. The Danish research ethics committee specifically waived the need for consent before collection of any data with the following statement, translated from Danish:*"it can be hereby announced that in connection with a questionnaire survey of 15-year-olds such as the one outlined, there is no requirement for informed consent (either oral or written) from parents/guardians".*

Data were treated anonymously throughout the process, and with a large study population, it is not possible to attribute results from the study to individuals. It is worth mentioning that the study only presents analyses and results that provide general knowledge about the study population and not knowledge about individual participants. The authors also had no access to information that could identify individual participants during or after data collection.

## Results

As can be seen in Table 1, women were more likely than men to report many depressive symptoms at ages 18 and 21. At age 28, the distribution was almost the same for women and men, with a tendency for the proportion of young men experiencing many depressive symptoms to be higher than the proportion of young women. A higher proportion of 15-year-olds with low SSS-school reported many depressive symptoms at age 18 and 21 compared to 15-year-olds with medium or high SSS-school. At age 28, an equalization of the difference between low, medium and high SSS-school was seen. The same trend holds for SSS-society, however the proportion with many depressive symptoms was lower for low SSS-society than for low SSS-school. For the variables income and educational level, no major differences were observed in the distribution of few and many depressive symptoms. Among 15-year-olds who experienced poor family functioning or severe bullying, the proportion reporting depressive symptoms was higher than among those who experienced none or light bullying or good family functioning at age 15.

The strongest correlation was between SSS-school and bullying with a Spearmans rho of -0.24. All other correlations were lower (result not shown).

Fig 2 shows the trajectories of depressive symptoms during adolescence and young adulthood related to the level of SSS-school at age 15. It shows that at age 15 there is at large gap between mean depressive symptoms in those with low SSS-school compared to those with medium or high SSS-school. The gab decreases over time, and at age 28 there is hardly any difference between the SSS-school groups. Fig 2 also illustrates that at age 18 a small increase in the level of depressive symptoms was seen across all three SSS-groups.

Table 2 presents the associations between SSS-school and depressive symptoms, unadjusted as well as adjusted. It shows that the odds of having many depressive symptoms at ages 18, 21 and 28 are higher for those with low SSS-school compared to those with high SSS-school. This is seen both for the crude estimates and for the adjusted estimates. The adjusted OR between low SSS-school and depressive symptoms at age 18 is OR 3.34 [1.84;6.08], at age 21 OR 3.31 [1.75;6.26] and at age 28 OR 2.12 [1.13;3.97]. The adjusted associations between medium SSS-school are not as strong, however statistically significant at age 18 and 28 compared to high SSS-school.

Equivalised household income and mothers' level of education, appear to have little or no association with depressive symptoms at ages 18, 21 and 28. Poor family functioning and severe bullying are both statistically significant associated with many depressive symptoms at all three age points.

## Discussion

This study documents that young people with low SSS-school at age 15 are more likely than young people with medium or high SSS-school at age 15 to report many depressive symptoms at age 18, 21 and 28. At age 28, the strength of the association decreases. The association was strongest for 18-year-olds.

Prior to the study, a thorough literature review was conducted. First, a scoping search and a broad search were performed. This was done to identify and narrow down the relevance of the

**Table 1. Distribution of exposure and co-variates in relation to depressive symptoms at age 18, age 21 and age 28.**

| | Depressive symptoms Age 18 (2007) | | | | Depressive symptoms Age 21 (2010) | | | | Depressive symptoms Age 28 (2017) | | | |
|---|---|---|---|---|---|---|---|---|---|---|---|---|
| | n | n | Few symptoms n (%) | Many symptoms n (%) | n | n | Few symptoms n (%) | Many symptoms n (%) | n | n | Few symptoms n (%) | Many symptoms n (%) |
| **Sex*** | 2156 | | | | 1789 | | | | 1719 | | | |
| Girls | | 1172 | 699 (60) | 473 (40) | | 1017 | 596 (59) | 421 (41) | | 709 | 544 (77) | 165 (23) |
| Boys | | 984 | 747 (76) | 237 (24) | | 772 | 511 (66) | 261 (34) | | 1010 | 745 (74) | 265 (26) |
| **SSS-school*** | 2095 | | | | 1745 | | | | 1679 | | | |
| Low | | 79 | 27 (34) | 52 (66) | | 68 | 23 (34) | 45 (66) | | 66 | 40 (61) | 26 (39) |
| Average | | 1086 | 709 (65) | 377 (35) | | 893 | 527 (60) | 356 (40) | | 873 | 643 (74) | 230 (26) |
| High | | 930 | 670 (72) | 260 (28) | | 784 | 523 (67) | 261 (33) | | 740 | 580 (78) | 160 (22) |
| **SSS-society*** | 2107 | | | | 1754 | | | | 1683 | | | |
| Low | | 33 | 18 (55) | 15 (45) | | 28 | 13 (46) | 15 (54) | | 28 | 20 (71) | 8 (29) |
| Average | | 1382 | 905 (65) | 477 (35) | | 1156 | 692 (60) | 464 (40) | | 1137 | 851 (75) | 286 (25) |
| High | | 692 | 484 (70) | 208 (30) | | 570 | 376 (66) | 194 (34) | | 518 | 392 (76) | 126 (24) |
| **Income**** | 2130 | | | | 1772 | | | | 1700 | | | |
| Lower | | 665 | 460 (69) | 205 (31) | | 538 | 314 (58) | 224 (42) | | 530 | 387 (73) | 143 (27) |
| Medium | | 715 | 467 (65) | 248 (35) | | 619 | 379 (61) | 240 (39) | | 590 | 447 (76) | 143 (24) |
| Higher | | 750 | 503 (67) | 247 (33) | | 615 | 400 (65) | 215 (35) | | 580 | 442 (76) | 138 (24) |
| **Mothers educational level**** | 2060 | | | | 1715 | | | | 1615 | | | |
| < 10 years | | 423 | 287 (68) | 136 (32) | | 347 | 188 (54) | 159 (46) | | 344 | 244 (71) | 100 (29) |
| 10–13 years | | 941 | 648 (69) | 293 (31) | | 773 | 496 (64) | 277 (36) | | 724 | 560 (77) | 164 (23) |
| 13–15 years | | 638 | 413 (65) | 225 (35) | | 541 | 342 (63) | 199 (37) | | 500 | 380 (76) | 120 (24) |
| > 15 years | | 58 | 37 (64) | 21 (36) | | 54 | 32 (59) | 22 (38) | | 47 | 28 (60) | 19 (40) |
| **Family functioning*** | 2067 | | | | 1724 | | | | 1643 | | | |
| Good | | 1577 | 1105 (70) | 472 (30) | | 1307 | 860 (66) | 447 (34) | | 1236 | 964 (78) | 272 (22) |
| Poor | | 490 | 285 (58) | 205 (42) | | 417 | 208 (50) | 209 (50) | | 407 | 274 (67) | 133 (33) |
| **Bullying*** | 2137 | | | | 1771 | | | | 1706 | | | |
| None | | 1596 | 1125 (70) | 471 (30) | | 1326 | 856 (65) | 470 (35) | | 1277 | 980 (77) | 297 (23) |
| Light | | 488 | 285 (58) | 203 (42) | | 410 | 227 (55) | 183 (45) | | 388 | 276 (71) | 112 (29) |
| Severe | | 53 | 26 (49) | 27 (51) | | 35 | 14 (40) | 21 (60) | | 41 | 24 (59) | 17 (41) |

* Information about sex, SSS-school, SSS-society, family functioning and bullying was assessed in 2004.

** Mean equivalised income, calculated by averaging the annual household income from the years when the study participants were 7–10 years old, categorised at 33.3rd and 66.6th percentile.

*** Mothers educational level assessed in 2004: < 10 years = primary/elementary school, 10–13 years = secondary school, 13–15 years = bachelor's degrees, > 15 years = master's and doctoral degrees.

problem. To our knowledge, this study is the first to prospectively examine the association between SSS in school and depressive symptoms in adolescence and early adulthood. However, the findings are in line with the findings from a study from 2020 by Poulsen et al. showing that objectively measured socioeconomic factors and SSS-society in early adolescence are associated with depressive symptoms in later adolescence and early adulthood [12]. However, Poulsen et al. found that the association disappear after adjusting for co-variates, which was not the case in this study where the association between SSS-school and depressive symptoms only weakened.

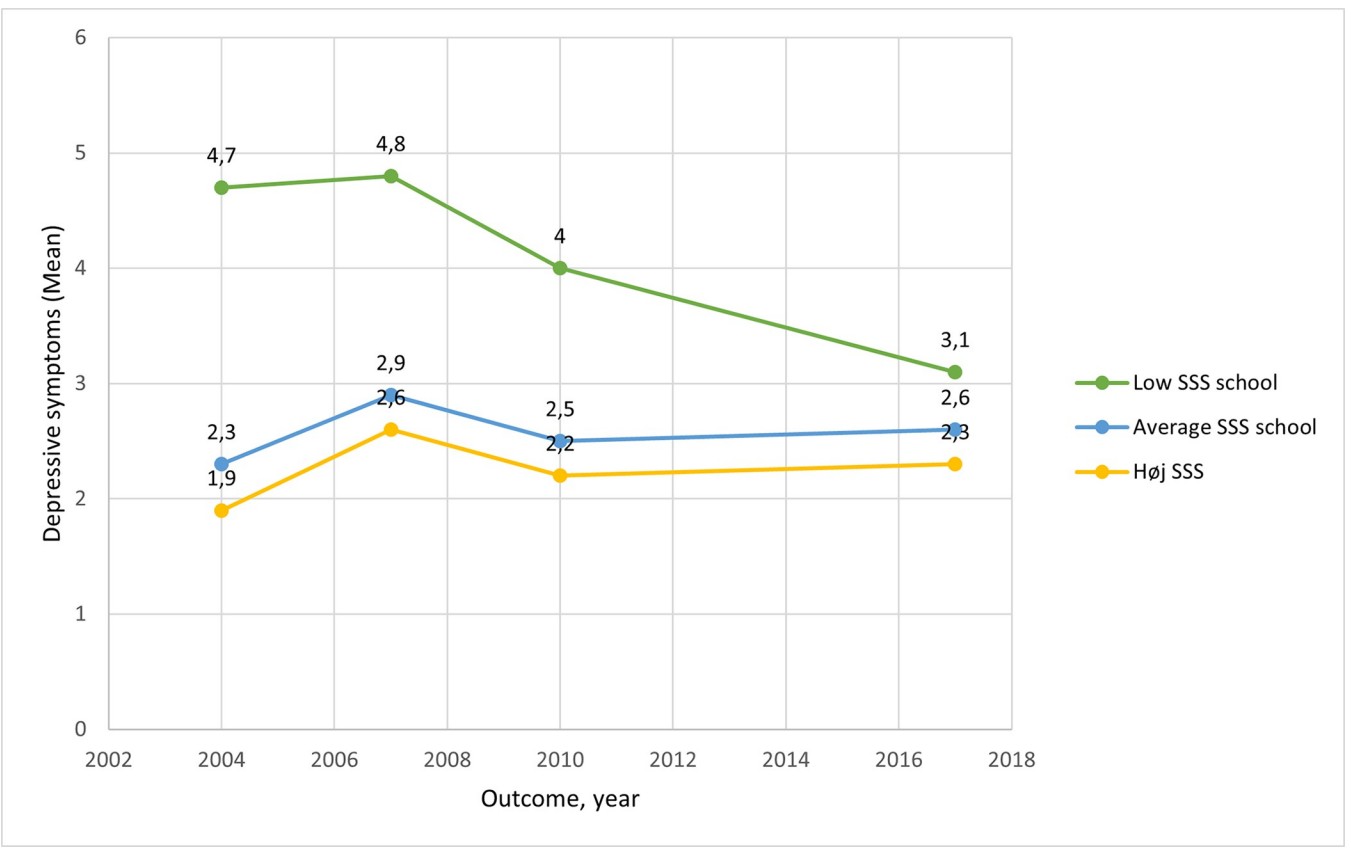

**Fig 2.**

In line with two cross-sectional studies by McLaughlin et al. and by Varga et al. [27, 29], this study finds that SSS-school is the social factor most strongly associated with depressive symptoms. However, McLaughlin et al. also found that the importance of SSS-school is only relevant if objective socioeconomic status is not low. This finding differs from the results of the present study, where additional analyses for the confounding effect of individual covariates, found that household income and maternal education level did not essentially confound the association.

In a Danish cross-sectional study from 2015, Nielsen et al. point out that the social capital of schools has a major impact on mental health among young people [50]. Similarly, a cohort study by Oldehinkel et al. found that social status in the classroom has an impact on depressive symptoms [51]. This fits well with this study's finding, where SSS-school measured among peers appears to have a greater impact on the odds of depressive symptoms than SSS in relation to society.

Cross-sectional studies by Pössel et al. from 2021 and by Quon et al. from 2015, showed that low SSS-school is associated with depressive symptoms [7, 28]. A meta-analysis by Quon et al., also showed that high SSS was associated with better mental health. Similarly, a point is made that objective socioeconomic factors may be difficult to apply when studying young people because they do not yet have an independent economy and therefore still are dependent on parental support [31]. This point is supported by our findings showing that SSS-school is a relevant factor in identifying adolescents with the highest odds of reporting multiple depressive symptoms as young adults.

**Table 2. The association between SSS, SES, family functioning and bullying at age 15, and many depressive symptoms at age 18, age 21 and age 28.**

| | 1. follow-up 2007, 18 years | | | | 2. follow-up 2010, 21 years | | | | 3. follow-up 2017, 28 years | | | |
|---|---|---|---|---|---|---|---|---|---|---|---|---|
| | n = | OR [95%CI] | n = | aOR [95%CI] | n = | OR [95%CI] | n = | aOR [95%CI] | n = | OR [95%CI] | n = | aOR [95%CI] |
| **SSS-school*** | 2095 | | 1880 | | 1745 | | 1576 | | 1679 | | 1472 | |
| Low | | 4.96 [3.05;8.07] | | 3.34 [1.84;6.08] | | 3.92 [2.32;6.62] | | 3.31 [1.75;6.26] | | 2.36 [1.40;3.98] | | 2.12 [1.13;3.97] |
| Average | | 1.47 [1.13;1.66] | | 1.26 [1.02;1.56] | | 1.33 [1.09;1.62] | | 1.20 [0.96;1.50] | | 1.30 [1.03;1.63] | | 1.30 [1.01;1.69] |
| High | | 1 (ref.) | | 1 (ref.) | | 1 (ref.) | | 1 (ref.) | | 1 (ref.) | | 1 (ref.) |
| **Sex*** | 2156 | | | | 1789 | | | | 1719 | | | |
| Boys | | 1 (ref.) | | | | 1 (ref.) | | | | 1 (ref.) | | |
| Girls | | 2.13 [1.77;2.57] | | | | 1.38 [1.14;1.68] | | | | 1.17 [0.94;1.47] | | |
| **SSS-society*** | 2107 | | | | 1754 | | | | 1683 | | | |
| Low | | 1.94 [0.96;3.92] | | | | 2.24 [1.04;4.79] | | | | 1.24 [0.54;2.89] | | |
| Average | | 1.23 [1.01;1.49] | | | | 1.30 [1.05;1.60] | | | | 1.05 [0.82;1.33] | | |
| High | | 1 (ref.) | | | | 1 (ref.) | | | | 1 (ref.) | | |
| **Income**** | 2130 | | | | 1772 | | | | 1700 | | | |
| Low | | 0.91 [0.73;1.14] | | | | 1.33 [1.05;1.68] | | | | 1.18 [0.90;1.55] | | |
| Medium | | 1.08 [0.87;1.34] | | | | 1.18 [0.93;1.49] | | | | 1.02 [0.78;1.34] | | |
| High | | 1 (ref.) | | | | 1 (ref.) | | | | 1 (ref.) | | |
| **Mothers educ. level**** | 2060 | | | | 1715 | | | | 1615 | | | |
| < 10 years | | 0.83 [0.47;1.48] | | | | 1.23 [0.69;2.20] | | | | 0.60 [0.32;1.13] | | |
| 10–13 years | | 0.80 [0.46;1.38] | | | | 0.81 [0.46;1.43] | | | | 0.43 [0.23;0.79] | | |
| 13–15 years | | 0.96 [0.55;1.68] | | | | 0.85 [0.48;1.50] | | | | 0.47 [0.25;0.86} | | |
| > 15 years | | 1 (ref.) | | | | 1 (ref.) | | | | 1 (ref.) | | |
| **Family functioning*** | 2067 | | | | 1724 | | | | 1643 | | | |
| Good | | 1 (ref.) | | | | 1 (ref.) | | | | 1 (ref.) | | |
| Poor | | 1.68 [1.37;2.08] | | | | 1.93 [1.55;2.42] | | | | 1.72 [1.34;2.20] | | |
| **Bullying*** | 2137 | | | | 1771 | | | | 1706 | | | |
| None | | 1 (ref.) | | | | 1 (ref.) | | | | 1 (ref.) | | |
| Light | | 1.70 [1.38;2.10] | | | | 1.47 [1.17;1.84] | | | | 1.34 [1.03;1.73] | | |
| Severe | | 2.48 [1.43;4.30] | | | | 2.73 [1.38;5.42] | | | | 2.34 [1.24;4.41] | | |

* Information about sex, SSS-school, SSS-society, family functioning and bullying was assessed in 2004.

** Mean equivalised income, calculated by averaging the annual household income from the years when the study participants were 7–10 years old, categorised at 33.3rd and 66.6th percentile.

*** Mothers educational level assessed in 2004: < 10 years = primary/elementary school, 10–13 years = secondary school, 13–15 years = bachelor's degrees, > 15 years = master's and doctoral degrees.

It is relevant to discuss whether the results of this study would be different if depressive symptoms at age 15 and stress were considered as confounders instead of intermediate factors for the association between SSS-school and depressive symptoms later in life. The location of these factors in relation to the time course is not irrelevant. In the present study, these factors were considered as a consequence of SSS-school based on the knowledge that low SSS-school can cause depressive symptoms and also can cause feelings of stress [13, 52, 53]. However, it is possible that a bi-directional association exists. Therefore, it is relevant to consider whether depressive symptoms could affect self-perceived social status, or if stress could influence the reporting of SSS-school. If depressive symptoms at age 15 and stress were instead treated as confounders and adjusted for in the analyses, this did not change the direction of the associations (results not shown).

Several studies have investigated SSS-society as a risk factor of depressive symptoms. This study investigated SSS-school which is a different perspective on the self-perceived social status. The results of our study confirms that SSS-school, may be an equally if not even more relevant measure of social status among young adolescents than the more societal perspective. The results indicate that the relationships at school are important for the way young people perceive themselves and their social status, and that this is important for the risk of developing depressive symptoms later in life. This finding is supported by a study from 2001 where Goodman et al. in a cross-sectional study examined the differences between school and society as assessed by the MacArthur Subjective Social Status Scale and found that SSS-school was more strongly associated with depressive symptoms than SSS-society [26].

To test the robustness of the study's findings, a sensitivity analysis was performed using alternative cut-points at the 90th percentile and at 3 points (out of 12 possible) on the CES-DC scale, respectively. Moreover, a complete-case analysis was performed. None of these complementary analyses changed the final conclusion. The results of this analysis can be provided upon request.

When examining the confounding effect from each covariate separately we found that bullying is an important confounder of the association. This is in line with a study by Yang et al. from 2021 showing that childhood trauma affects the association between socioeconomic status and depressive symptoms [9]. These two findings are well aligned as bullying definitely should be seen as a significant childhood trauma [54].

## Strength and limitations

A strength of the study is that the longitudinal design allowed us to investigate the association between SSS-school and depressive symptoms over both a short and a long time period. The fact that exposure and co-variates were measured prior to outcome, increases the possibility to investigate a causal relation. Similarly, the large study population and the relatively high response rate at the initial questionnaire collection (83% response rate) are strengths of the study.

The use of self-reported data is seen as an opportunity to examine the participant's own assessment of their self-perceived social status and symptoms of depression.

Another strength of the study is the use of almost complete register-based data to identify the objective socio-economic perspective, which reduces the risk of information bias. Furthermore, exposure and outcome data were obtained from questionnaires using validated measurement tools, respectively the MacArthur scale to assess SSS [26] and the CES-DC/D scale to assess depressive symptoms [44]. The use of validated tools increases the internal validity of the study and allows for better precision and high validity. However, the study uses only one fifth of the questions from the original CES-DC scale, why the risk of misclassification of

depressive symptoms is present. Additionally, it is important to note that the study examines symptoms of depression and not clinical depression.

Non-participation can lead to selection bias if characteristics of the participants in the study differ significantly from characteristics of the non-participants. The dropout analysis showed that non-participants were more likely to be boys, to come from families with low household income, to have a low educated mother and poor family functioning (results not shown). A study by Winding et al. on the same cohort showed that non-participation and loss to follow-up from the West Jutland Cohort study did not have any significant impact on the relative risk measures, which is reassuring for the results of this study [38].

## Conclusions

In this Danish prospective cohort study, statistically significant associations between low SSS-school at age 15, measured in relation to peers, and the odds of experiencing many depressive symptoms at ages 18, 21 and 28 were found. Associations were also found between medium SSS-school and the odds of experiencing many depressive symptoms, however, these associations were not nearly as strong. The associations were adjusted for relevant co-variates including objectively measured socioeconomic factors. The association decrease with increasing age, suggesting that SSS-school at age 15 has the greatest impact on the prevalence of many depressive symptoms in the short term, and fades over time.

To reduce the risk of depressive symptoms in young adulthood it is relevant to be aware of the adolescents subjective assessed social status at school, and particular attention should be paid to those who rate their subjective social status as low.

## Supporting information

**S1 Checklist. STROBE statement—checklist of items that should be included in reports of observational studies.**
(DOCX)

## Acknowledgments

For supervision and professional support the author wants to give a special thanks to Trine Nøhr Winding and Vivi Just-Nørregaard.

## Authors' information

Marie Kjærgaard Lange, cand.scient.san., Department of Public Health Aarhus University, e-mail: marieklange@gmail.com.

## Author Contributions

**Writing – original draft:** Marie Kjærgaard Lange.

**Writing – review & editing:** Vivi Just-Nørregaard, Trine Nøhr Winding.

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
