## [Decision Letter · Decision Letter 0]

16 Aug 2023

PONE-D-23-19172How does subjective social status at age 15 affect the risk of depressive symptoms in young adulthood? A longitudinal studyPLOS ONE

Dear Dr. Lange,

Thank you for submitting your manuscript to PLOS ONE. After careful consideration, we feel that it has merit but does not fully meet PLOS ONE’s publication criteria as it currently stands. Therefore, we invite you to submit a revised version of the manuscript that addresses the points raised during the review process.

An expert in this field has carefully reviewed this submission and made detailed suggestions. I am not sure if the authors could satisfactorily address all the comments; however, I would like to give a chance for the authors to resubmit. 

We look forward to receiving your revised manuscript.

Kind regards,

Chung-Ying Lin

Academic Editor

PLOS ONE

Reviewers' comments:

Reviewer's Responses to Questions

**Comments to the Author**

1. Is the manuscript technically sound, and do the data support the conclusions?

Reviewer #1: Partly

2. Has the statistical analysis been performed appropriately and rigorously? 

Reviewer #1: Yes

3. Have the authors made all data underlying the findings in their manuscript fully available?

Reviewer #1: No

4. Is the manuscript presented in an intelligible fashion and written in standard English?

Reviewer #1: No

5. Review Comments to the Author

Reviewer #1: Dear authors: thank you for allowing me to review your manuscript. It’s also an excellent opportunity to learn from your work and culture. Please take the comments as a kind encouragement, and hope it helps.

1. The authors may want to define the studied time points. Why the time points were chosen? I understand that the study used secondary data, but why the periods were explored still needs to be explained. With well-defined time points, the references could better support the following statements.

2. Line 64: “The transition from adolescence to adulthood is an important developmental life period in which many physical, psychological as well as social changes occur, and a large number of young people struggle with mental health problems in this particular life phase.” The statement may need to be justified that why changes will bring health problems and consequences for health. Understandably, depression could be the problem that developmental challenges lead to, but the processing or transition needs to be clarified and logicalized.

3. Line 74-75:＂Social status is often assessed as objective socio-economic status (SES), with classification based on information about income, educational level and/or labor market participation.” The statement is oversimplified with respect to the definition of social status. There are several categorizing methods with their respective concepts. Please provide references to support the categorizing method used in this study.

4. Line 78-80: “Several studies have found low SSS in adulthood to be linked to mental health conditions including depression and depressive symptoms”. Some concern here, the statement is oversimplified how depression or other mental health conditions cause. Please explain the association between SSS and depression/mental conditions to solidify the structure of the manuscript and the following analyses.

5. Line 80-83: “In young people SSS can be measured in two ways; first as SSS-society which is the self-rated social status the young person assesses his/her family to have compared to other families in the society, and, second as SSS-school which is the young person's self-rated social status compared to the peers at school.” What is the difference between these two measures? Please explain the meaning and impact of these two measures on adolescents with references to support and rationalize the SSS used in the study.

6. Line 84-86. “Social interactions are important for young people and young people assess their own status and worth in relation to others. Many hours are spent at school and in Denmark 98% of children are enrolled in either private or primary school.” These sentences are not logical. Please re-organize the sentences.

7. Line 87-90: “A cross-sectional study by Goodman et al. showed that the average school income, has a positive impact on reducing adolescents' depressive symptoms.” Why school income? It was mentioned that SSS is based on information about income, educational level, and/or labor market participation”. It seems like income is not simply the only factor of SSS. Also, what “school” income? Could the statements and references limit the meaning of SSS?

8. The study title is “How does subjective social status at age 15 affect the risk of depressive 4 symptoms in young adulthood? A longitudinal study”; however, “the aim of this study is to investigate the association between SSS-school at age 15 and the risk of developing depressive symptoms at ages 18, 21 and 28.” They don’t seem consistent. To unify the study’s variables, the authors may want to clarify the confusion.

9. The authors may want to define “15-year-old's subjective social status” as social status in school, with related and supportive references for the present study. However, the definition is unclear, and its meaning and the association with potential mental conditions are unfound in the introduction section, which could confuse the structure of the study.

10. Why the authors focused on depression in this study? The authors’ motivation, the research background, and the associations between variables were not explained enough. The logic of the study was not recognized in the study.

11. Line 121: Since the data was secondary, the flowchart of how the participants were recruited is not proper here.

12. Line 133-135: “Each item was a description of a feeling for which the respondent had to indicate the degree to which it applied to him/her. The degree of agreement could be indicated as "Not at all", "A little", "Sometimes" or "A lot".” How many points do these items represent?

13. Line 137: “In this study, participants could score from 0-12 by answering the four questions.” What are the 4 questions? How to do the scaling? Still indicated from "Not at all", "A little", "Sometimes" or "A lot"? Do they scale from 0-3? Please clarify the scaling method.

14. Why the SSS-school and SSS-society were categorized and divided into three categories? What is the principle?

15. Line 168-169: Why was the objective socioeconomic status measured by the mean equivalised income and mothers’ educational level? Please provide the principle and references to explain the usage.

16. Line 189: “Bullying” is coming from nowhere and hasn’t been mentioned in the previous statements. Is it related to depression?

17. Line 194: Why stress and BMI are here?

18. The “few” versus “many” depressive symptoms seem vague. Authors may want to re-dine the term of the compared set.

19. The multiple logistic regression needs more details while describing the analytic processes.

20. In the discussion: “This study is the first to prospectively examine the association between SSS in school and depressive symptoms in adolescence and early adulthood.” The study has processed a great idea by using the data to reveal the phenomenon. However, the research background and literature review must be significantly strengthened to justify and solidify the research concept and structure.

21. “Several studies have investigated SSS-society as a risk factor of depressive symptoms. This study investigated SSS-school which is a different perspective on the self-perceived social status. The results of our study confirm that SSS-school, may be a more relevant measure of social status among young adolescents than the more societal perspective.” It is a great concept, but no substantial evidence/references support it. Unfortunately, such a problem often happens in this study: Nice perspectives are presented, but they are narrations because of no solid references to support them. Therefore, proper references are strongly encouraged.

22. Professional editing is recommended.

6. PLOS authors have the option to publish the peer review history of their article (what does this mean?). If published, this will include your full peer review and any attached files.

Reviewer #1: No

---

## [Author Response · Author response to Decision Letter 0]

6 Dec 2023

Thank you for the valuable response and the opportunity to revise our manuscript. We have gone through the review comments by the reviewer and implemented them in the revised manuscript. Please find the manuscript with tracked changes attached as a separate file. In addition, a clean copy of the revised manuscript is attached. Below you will find replies to each of the questions point-by-point (line numbering refers to the revised version of the manuscript with tracked changes).

---

## [Editor Report · Decision Letter 1]

12 Dec 2023

How does subjective social status at school at the age of 15 affect the risk of depressive symptoms at the ages of 18, 21, and 28? A longitudinal study

PONE-D-23-19172R1

Dear Dr. Lange,

We’re pleased to inform you that your manuscript has been judged scientifically suitable for publication and will be formally accepted for publication once it meets all outstanding technical requirements.

Kind regards,

Chung-Ying Lin

Academic Editor

PLOS ONE

Additional Editor Comments (optional):

The original reviewer was too busy to re-review the manuscript. Therefore, I have read the revision together with the response letter. I feel that the authors have satisfactorily addressed all the concerns from the previous reviewer. Therefore, I am glad to accept this paper. 
---

## [Editor Report · Acceptance letter]

18 Dec 2023

PONE-D-23-19172R1 

PLOS ONE

Dear Dr. Lange, 

I'm pleased to inform you that your manuscript has been deemed suitable for publication in PLOS ONE. Congratulations! Your manuscript is now being handed over to our production team.

Kind regards, 

on behalf of

Dr. Chung-Ying Lin 

Academic Editor

PLOS ONE